# Globally Disseminated Multidrug Resistance Plasmids Revealed by Complete Assembly of Multidrug Resistant *Escherichia coli* and *Klebsiella pneumoniae* Genomes from Diarrheal Disease in Botswana

Teddie O. Rahube [1,2] , Andrew D. S. Cameron [2,3,*] , Nicole A. Lerminiaux [2,3] , Supriya V. Bhat [2,3] and Kathleen A. Alexander [4,5]

1   Department of Biological Sciences and Biotechnology, Faculty of Science, Botswana International University of Science and Technology (BIUST), Private Bag 16, Palapye, Botswana
2   Institute for Microbial Systems and Society, Faculty of Science, University of Regina, Regina, SK S4S 0A2, Canada
3   Department of Biology, Faculty of Science, University of Regina, Regina, SK S4S 0A2, Canada
4   Department of Fish and Wildlife Conservation, Virginia Tech, Blacksburg, VA 24061, USA
5   Centre for African Resources: Animals, Communities and Land Use, Kasane, Botswana
*   Correspondence: andrew.cameron@uregina.ca; Tel.: +1-(306)-337-2568

**Abstract:** Antimicrobial resistance is a disseminated global health challenge because many of the genes that cause resistance can transfer horizontally between bacteria. Despite the central role of extrachromosomal DNA elements called plasmids in driving the spread of resistance, the detection and surveillance of plasmids remains a significant barrier in molecular epidemiology. We assessed two DNA sequencing platforms alone and in combination for laboratory diagnostics in Botswana by annotating antibiotic resistance genes and plasmids in extensively drug resistant bacteria from diarrhea in Botswana. Long-read Nanopore DNA sequencing and high accuracy basecalling effectively estimated the architecture and gene content of three plasmids in *Escherichia coli* HUM3355 and two plasmids in *Klebsiella pneumoniae* HUM7199. Polishing the assemblies with Illumina reads increased base calling precision with small improvements to gene prediction. All five plasmids encoded one or more antibiotic resistance genes, usually within gene islands containing multiple antibiotic and metal resistance genes, and four plasmids encoded genes associated with conjugative transfer. Two plasmids were almost identical to antibiotic resistance plasmids sequenced in Europe and North America from human infection and a pig farm. These One Health connections demonstrate how low-, middle-, and high-income countries collectively benefit from increased whole genome sequencing capacity for surveillance and tracking of infectious diseases and antibiotic resistance genes that can transfer between animal hosts and move across continents.

**Keywords:** antimicrobial resistance; next generation sequencing; plasmids; antibiotic resistance genes; Botswana

## 1. Introduction

Antimicrobial resistance (AMR) threatens the future availability of therapies to cure and prevent infectious diseases. Antimicrobial treatment failures result in increased health burdens and medical costs, elevating morbidity and mortality rates. Of critical concern is the declining effectiveness of medications to treat bacterial infections, especially in low- and middle- income (LIMCs) countries where bacterial disease burdens are greatest, and antibiotics can help protect child and maternal health. Bacterial resistance to antibiotics is estimated to become the greatest challenge in healthcare, with Africa forecast to suffer the largest negative impacts [1,2].

A driving force of the resistance pandemic is the ability of many antibiotic resistance genes (ARGs) to transfer between bacteria [3]. The high rates and facility with which these 'acquired' resistance genes mobilize between bacteria make them moving targets that do not conform to standard epidemiological tracking or respond to classic interventions. Genetic elements called plasmids are primary vectors for ARG mobility. Because plasmids replicate independently of the bacterial chromosome, they can be inherited vertically by daughter cells or can be transferred horizontally to adjacent cells in bacterial communities [3]. Prime examples include the carbapenem-resistant and extended-spectrum ß-lactamase encoding plasmids in Enterobacteriaceae that spread globally across clinical and non-clinical environments soon after first detection [4]. Another important feature of ARGs is that they can coalesce into gene islands. Subsequent acquisition of a multi-resistance island by a bacterium contributes directly to the emergence of multi-drug resistant cell.

The genetic composition of plasmids changes through high rates of recombination that add, subtract, and shuffle genes. The plasticity of plasmid DNA sequences, the modularity of gene cassettes, and the abundance of repetitive elements in plasmids hampers the assembly of whole plasmids from short-read whole genome and metagenomic datasets [5,6]. Consequently, plasmid epidemiology is poorly understood. Long-read DNA sequencing platforms are making strides towards assembling whole plasmids for genomic epidemiology [7], but the most accessible platforms from Oxford Nanopore are still hindered by lower quality of base calling.

Characterizing and tracking plasmids is a necessary step in the global effort to identify, track, and mitigate the spread of AMR. Whole genome sequencing of bacterial isolates enhances AMR surveillance at national and international scales, and has been successfully applied in LMICs to identify drivers of AMR [8–11]. Botswana is a developing country in southern Africa with high prevalence and burdens of communicable diseases [12]. Current infectious disease diagnostics in Botswana rely primarily on traditional culture-based methods for isolation and characterization of infectious bacteria. Studies of antibiotic resistance in Botswana rely on conventional microbiological culture and PCR methods to isolate antibiotic resistant bacteria and detect ARGs [13–15]. Cost and training barriers have impaired deployment of next generation DNA sequencing in LMICs. These barriers include limited availability of sequencing platforms, a lack of standardized definitions and quality control metrics, and a shortage of computer capacity [16]. In Botswana for example, DNA sequencing is contracted to South African service labs. An exception is the use of Oxford Nanopore sequencing at the Botswana-Harvard AIDS Institute Partnership (BHP) reference laboratory, but this is solely for HIV genotyping. Consequently, Botswana researchers rely on international collaborations to conduct genome sequencing projects while Botswana public health lacks access to next generation DNA sequencing.

The primary objective of this study was to compare assembly and annotation workflows using Illumina and Oxford Nanopore sequence data for accurate identification and characterization of plasmids and ARGs. We also wished to address the deficit of sequenced and assembled multidrug resistance plasmids in Africa. The study was conducted using multidrug resistant bacterial isolates from human infections in the Chobe district in Northern Botswana, which is globally recognized as a wildlife tourist destination that attracts many international visitors. From our assessment of the cost and effectiveness of DNA sequencing methodologies for deployment in LMICs, we present the benefits of implementing DNA sequencing for identification of antibiotic resistant bacteria and their plasmids in Botswana.

## 2. Materials and Methods

### 2.1. Bacterial Isolation and Antibiotic Susceptibility Testing

Two extensively multi-drug resistant bacterial isolates were acquired from patients hospitalized for diarrheal disease in Chobe District, Botswana in 2012. These fecal samples were collected as part of a long-term study of diarrheal disease and antimicrobial resistance in Chobe District [17–20]. This study was conducted under permit from the Botswana

Ministry of Environment, Natural Resources Conservation and Tourism (EWT8/36/4) and the Botswana Ministry of Health and Wellness (HPSME:13/18/1 Vol. X [878]). Approval was also obtained from the Virginia Tech Institutional Review Board (#11–573). Fecal samples used in this study represent archived, anonymized laboratory samples collected from patients presenting to the primary hospital and clinics in Chobe District, Botswana. All study activities were carried out in accordance with relevant guidelines and regulations.

Bacterial isolates were obtained from both human faecal samples and tested for antibiotic resistance against antibiotic classes commonly used in Botswana [13]: ampicillin (10 µg/mL; AMP10; penicillin), ceftiofur (30 µg/mL; XNL30; cephalosporin), chloramphenicol (30 µg/mL; C30; aminoglycoside), ciprofloxacin (5 µg/mL; CIP5; flouroquinolone), doxycycline (30 µg/mL; D30; tetracycline), gentamycin (10 µg/mL; G10; aminoglycoside), neomycin (30 µg/mL; N30; aminoglycoside), streptomycin (10 µg/mL; S10; aminoglycoside), sulfamethoxazole-trimethoprim (25 µg/mL; SXT25; sulfonamide), and tetracycline (30 µg/mL; T30; tetracycline) (BBL, Becton Dickinson Company) using the Kirby-Bauer disk diffusion method as previously described [13].

## 2.2. Whole Genome Sequencing

Genomic DNA was extracted using a BioBasic EZ-10 spin column genomic DNA miniprep kit (BS423). Short-read library preparation was conducted using the NEBNext Ultra II DNA library kit for Illumina (E7645) and the NEBNext® Multiplex Oligos for Illumina (E7335). All steps were conducted according to the NEB protocols. HUM3355 was sequenced on the Illumina MiSeq platform using Reagent Kit v2 (300-cycle, Illumina Inc.), producing 2 × 150 bp paired-end reads. HUM7199 was sequenced on the MiSeq platform using Reagent Kit v3 (600-cycle, Illumina Inc., San Diego, CA, USA), producing 2 × 300 bp paired-end reads.

For long-read sequencing, the HUM3355 DNA was sheared using g-TUBEs (Covaris Inc., Woburn, MA, USA) following the manufacturer's protocol to obtain fragments averaging around 8 kbp in length, whereas the HUM7199 DNA was not sheared by g-TUBE. Long-read sequencing libraries were prepared using the SQK-LSK109 ligation sequencing kit and the EXP-NBD103 native barcoding expansion kit (Oxford Nanopore Technologies, Oxford, UK). Libraries were sequenced on a MinION R9.4.1 flow cell. All library preparation and sequencing steps were conducted according to the Oxford Nanopore Technologies protocols. Basecalling was performed by Guppy v4.2.2 using the HAC model (dna_r9.4.1_450bps_hac.cfg), followed by demultiplexing.

## 2.3. Genome Assemblies

For Illumina sequencing reads, we used Trimmomatic v0.35 [21] to remove adapter sequences and eliminated reads with an average Qscore <30 across a 4-bp sliding window. After quality trimming and filtering, 98.6 Mbp (HUM3355) and 69.4 Mbp (HUM7199) provided 19-fold and 13-fold genome coverage, respectively. Paired and unpaired reads were assembled with Unicycler v0.4.7 [22], including polishing via Pilon v1.23 [23]. For Nanopore sequencing reads, we used Porechop v0.2.3 (https://github.com/rrwick/Porechop, accessed on 26 April 2021) to remove barcodes and adaptors, and Filtlong v0.2.0 (https://github.com/rrwick/Filtlong, accessed on 26 April 2021) to eliminate reads with a mean quality <90 and a read length <1 kb. After quality filtering, average read length was 4048 bp for HUM3355 and 8840 bp for HUM7199, totalling 709 Mbp (140-fold coverage) and 3800 Mbp (700-fold coverage), respectively. We used Trycycler v0.4.1 [24] to generate a consensus long-read assembly from 50X subsampled read sets assembled by Flye v2.8.1 [25], Raven v1.3.0 [26], Canu v2.1.1 [27] and miniasm/Minipolish v0.3/v0.1.2 [28]. Assemblies were polished with Medaka v1.2.1 (https://github.com/nanoporetech/medaka, accessed on 26 April 2021) and Pilon v1.23 [23]. Genomes were annotated with the NCBI Prokaryotic Genome Annotation Pipeline (PGAP) [29] (Figure S1).

*2.4. Sequence Analysis, Visualization, and Annotation*

The two multi-drug resistant bacterial isolates were identified to species level by submitting the genomes assembled from Illumina sequencing to the Public databases for molecular typing and microbial genome diversity (PubMLST; https://pubmlst.org, accessed on 1 July 2019) [30]. *De novo* assembled whole genome sequences were visualised using Bandage [31].

Plasmid assembly alignments were created with BRIG [32]. The whole genome assembled contigs were imported into the Pathosystems Resource Integration Center, PATRIC (https://patricbrc.org/, accessed on 26 July 2021) for comprehensive genome analysis [33]. Plasmid contigs from polished assemblies were annotated with the Rapid Annotation for Subsystem Technology (RAST) server [34]. Resistance gene identifier (RGI) software on the Comprehensive Antibiotic Resistance Database (CARD) was used to predict and analyze the resistome of each whole genome assembly [35]. Identification of acquired ARGs was conducted in ResFinder database [36]. Replicon typing was conducted using PlasmidFinder server available online at the Center for Genomic Epidemiology (http://www.genomicepidemiology.org/services/, accessed on 1 July 2019) [37]. MOB groups were predicted by MOBScan [38], and plasmid taxonomic units were assigned by COPLA [39]. Visual annotated physical maps of plasmids were generated using the BLAST ring image generator [32] and Vector NTI 10.3.0 (Invitrogen Corporation, Carlsbad, CA, USA).

## 3. Results

*3.1. Comparison of Illumina and Nanopore DNA Sequence Data for Whole Genome Assemblies*

Whole genome sequencing was conducted for two multi-drug resistant bacterial isolates, HUM3355 and HUM7199, that were isolated from human diarrhoea. Quality trimmed read data from Illumina (short-read) and Nanopore (long-read) platforms were assembled using dedicated workflows optimized for each type of read data (Figure S1). The Public databases for molecular typing and microbial genome diversity (PubMLST) designated HUM3355 as *Escherichia coli* and HUM7199 *Klebsiella pneumoniae*.

Visualization of the de novo genome assemblies was conducted using Bandage (31). Assemblies from the Nanopore data resolved singular circular chromosomes and multiple extrachromosomal elements in both species (Figure 1A). Assembly of Nanopore reads predicted a 4,714,922 bp chromosome plus four circular extrachromosomal elements in HUM3355, though the smallest contig was discarded as spurious because not a single read in the in the Illumina dataset matched sequence in this contig. Assembly of Nanopore reads predicted a 5,106,117 bp chromosome plus two circular extrachromosomal elements in HUM7199. Assemblies based on Illumina-only data predicted genome lengths of 4,859,045 bp for *E. coli* HUM3355 and 5,395,117 bp for *K. pneumoniae* HUM7199, fragmented in 250 and 2149 contigs, respectively, with a low number of potential connections between contigs in both genomes (Figure 1B).

The higher error rates of third generation (long-read) DNA sequencing can be corrected by polishing a long-read assembly using higher accuracy second-generation (short-read) DNA sequencing data from the same genome. Pilon is a program that uses short-read datasets to polish assemblies by correcting bases and fixing small-scale mis-assemblies [23]. Polishing the Nanopore assemblies with the Illumina data had only minor impact on the overall assemblies and annotations (Figure 1A); the bioinformatic workflow is illustrated in Figure S1. Hybrid polishing the HUM3355 genome made only very slight changes to the estimated sizes of replicons. Hybrid polishing of the HUM7199 genome reduced the overall estimated size by 3656 bp (less than 0.07%), yet retained the three replicon architecture resolved by Nanopore alone.

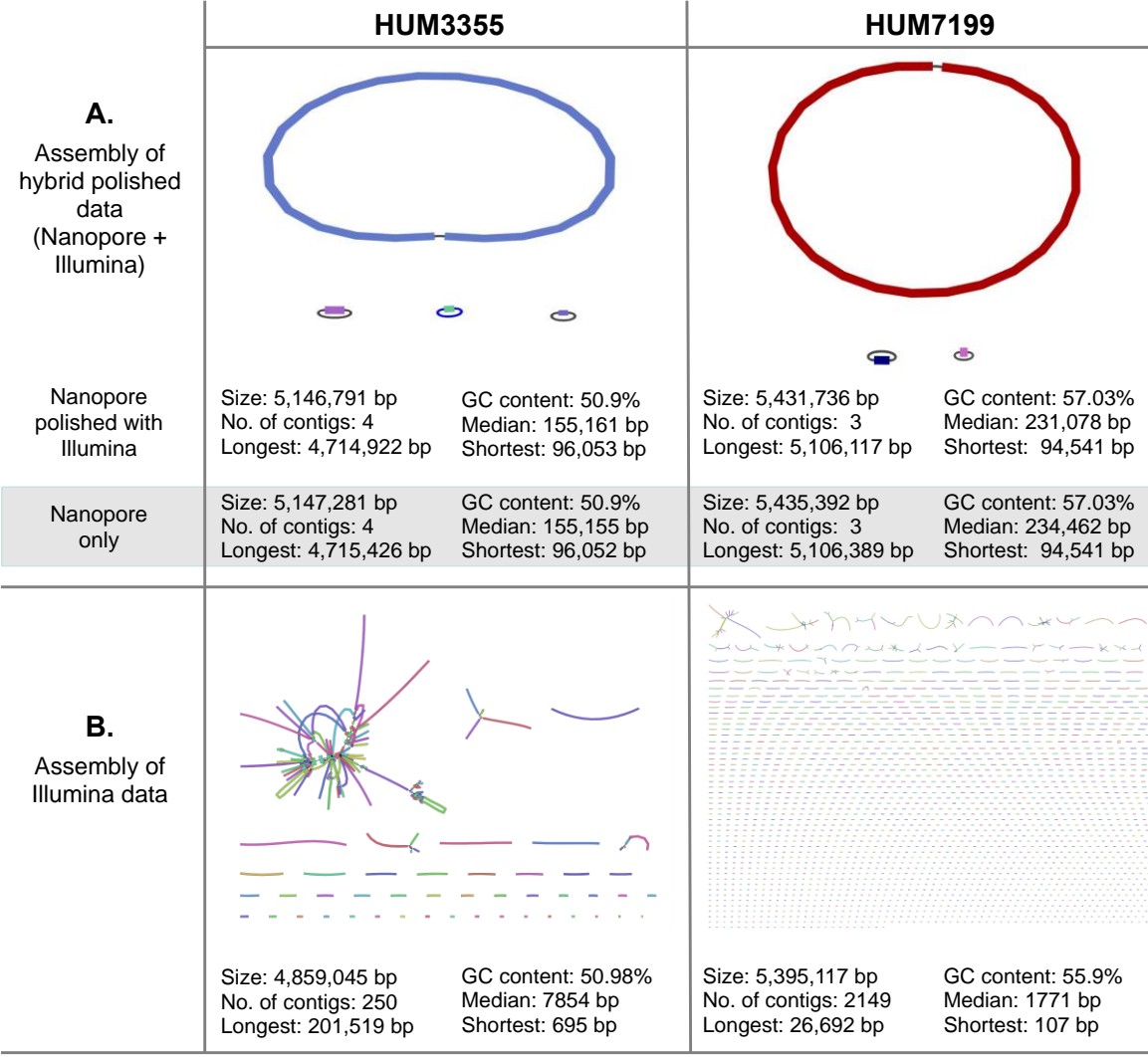

**Figure 1.** Visualization and comparison of whole genome assemblies for *E. coli* HUM3355 and *K. pneumoniae* HUM7199. (**A**) Contigs assembled from Oxford Nanopore Technologies MinION sequencing polished with Illumina reads. Assembly metrics are provided for the polished genomes and the Nanopore assemblies pre-polishing by Pilon. (**B**) Contigs assembled from Illumina sequencing data. Bandage [31] was used to visualize assemblies.

### 3.2. Whole Genome Comparative Analysis and Detection of Antibiotic Resistance Genes

Oxford Nanopore Technologies sequencing has the distinct advantage of being able to resolve plasmids in bacterial genomes. However, historically lower basecalling accuracy raises concerns about the quality of variant identification and gene annotations. We compared the Comprehensive Antibiotic Resistance Database (CARD) predictions of ARGs in the hybrid polished, Nanopore only, and Illumina only genome assemblies. There was high agreement between the hybrid polished and Nanopore only assemblies (Figure 2). In the hybrid polished *E. coli* HUM3355 genome, CARD identified 40 perfect hits, 36 of which were also identified in the Nanopore only assembly. CARD also identified 36 of the same genes in the Illumina only assembly, but only 27 as perfect matches to the CARD references. CARD identified 13 perfect ARG hits in the polished and Nanopore only HUM7199 assemblies (Figure 2), three of which were not detected in the Illumina only assembly.

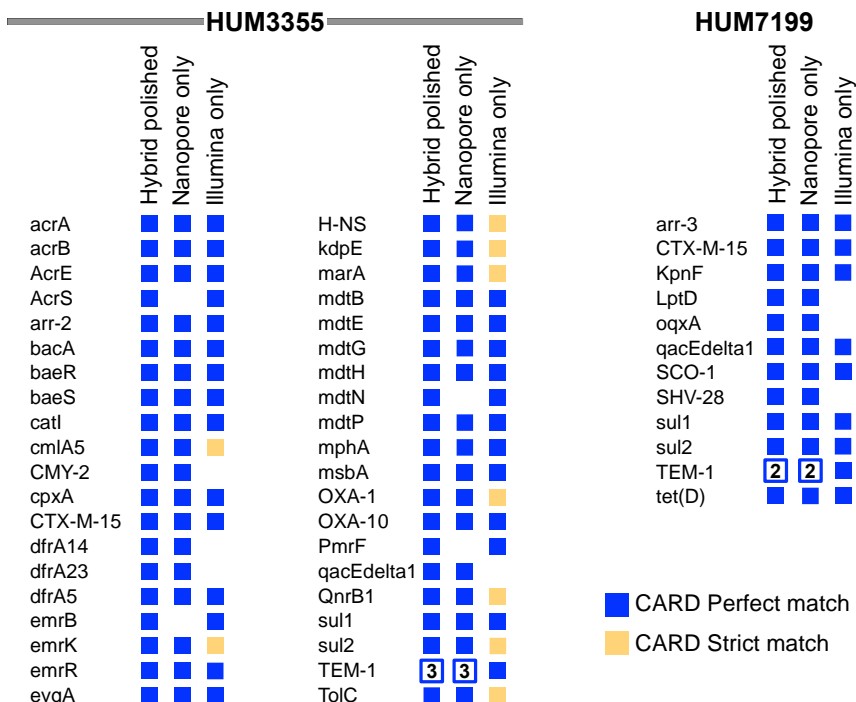

**Figure 2.** Antimicrobial resistance genes identified by CARD in HUM3355 and HUM7199 whole genome assemblies. Perfect matches to reference genes in the hybrid polished assemblies are compared to the Nanopore only and Illumina only assemblies. Genes are listed alphabetically using their CARD short names. An empty space indicates that the gene was not identified as a perfect or strict match in a genome assembly. The number of copies of beta-lactamase TEM-1 are indicated in open squares.

The *E. coli* HUM3355 and *K. pneumoniae* HUM7199 isolates have similar AMR phenotypes (Tables 1 and 2), but quite different profiles of acquired ARGs and chromosomal mutations predicted to confer resistance (Figure 2). The acquired ARGs belong to at least nine different classes; aminoglycoside (*aac(3)-IIe, acc(6)-Ib, ant(3)-IIa, aadA16, aph(3)-Ia, aph(3)-Ib, aph(6)-Id*), beta-lactam (*bla_{CMY-42}, bla_{CTX-M-15}, bla_{OXA-1}, bla_{OXA-10}, bla_{SCO-1}, bla_{SHV-18}, bla_{TEM-1β}*), fluoroquinolone (*qnrB1, oqxA*), fosfomycin (*fosA6*), folate-pathway antagonist (*drfA5, drfA14, drfA23, sul1, sul2*), macrolide (*mph(A)*), phenicol (*catI, catA1, cmlA5*), rifamycin (*arr-2, arr-3*), tetracycline (*tet(A), tet(B), tet(D)*), and quaternary ammonium compounds (*qacEdelta*).

**Table 1.** HUM3355 whole genome analysis of antibiotic resistance phenotypes and predicted antimicrobial resistance genes.

| Antimicrobial Class | Antibiotic Disc * (Phenotype) | | Gene Predicted to Confer Resistance ** | | | |
|---|---|---|---|---|---|---|
| | | | Chromosome (4,714,316 bp) | pHUM3355_A (168,684 bp) | pHUM3355_B (155,161 bp) | pHUM3355_C (96,053 bp) |
| Aminoglycoside | G10 | (R) | *aph(3″)-Ib,* | *aadA1,* | | |
| | N30 | (R) | *aph(6)-Id,* | *aph(3″)-Ib,* | *aph(6)-Id* | — |
| | S10 | (R) | *mdf(A)* | *aph(6)-Id,* | | |
| | | | | *aac(3)-IIa,* | | |
| | | | | *aac(6′)-Ib-cr* | | |
| Beta-lactam | AMP10 | (R) | *bla_{TEM-1β}* | *bla_{TEM-1β}* | *bla_{TEM-1β}* | *bla_{CMY-2}* |
| | XNL30 | (R) | | *bla_{CTX-M-15},* | | |
| | | | | *bla_{OXA1},* | | |
| | | | | *bla_{OXA10}* | | |

**Table 1.** *Cont.*

| Antimicrobial Class | Antibiotic Disc * (Phenotype) | | Gene Predicted to Confer Resistance ** | | | |
|---|---|---|---|---|---|---|
| | | | Chromosome (4,714,316 bp) | pHUM3355_A (168,684 bp) | pHUM3355_B (155,161 bp) | pHUM3355_C (96,053 bp) |
| Folate pathway antagonist | SXT25 | (R) | *drfA14*, *sul2* | *drfA23*, *sul1*, *sul2* | *drfA5*, *sul2* | — |
| Fluoroquinolone | CIP5 | (S) | *mdf(A)* | *qnrB1*, *aac(6′)-Ib-cr* | — | — |
| Macrolide | not tested | | *mph(A)*, *mdf(A)* | *mac(A)*, *mac(B)* | — | — |
| Phenicol | C30 | (R) | *catA1*, *mdf(A)* | *catB3*, *cmlA1* | — | — |
| Rifamycin | not tested | | — | *arr2* | — | — |
| Tetracycline | D30 T30 | (R) (R) | *tet(B)*, *mdf(A)* | *tet(A)* | *tet(A)* | — |
| Quaternary ammonium compounds | not tested | | — | *qacEDelta* | — | — |

\* Antibiotics Disc diffusion panel: G10, gentamycin; N30, neomycin; S10, streptomycin; AMP10, ampicillin; XNL30, ceftiofur; SXT25, sulfamethoxazole-trimethoprim; CIP5, ciprofloxacin; C30, chloramphenicol; D30, doxycycline; T30, tetracycline; —, No predicted resistance gene detected. \*\* ARGs predicted by ResFinder and CARD (perfect and strict).

**Table 2.** HUM7199 whole genome whole genome analysis of antibiotic resistance phenotypes and predicted antimicrobial resistance genes.

| Antimicrobial Class | Antibiotic Disc * (Phenotype) | | Gene Predicted to Confer Resistance ** | | |
|---|---|---|---|---|---|
| | | | Chromosome (5,105,496 bp) | pHUM7199_A (231,078 bp) | pHUM7199_B (94,541 bp) |
| Aminoglycoside | G10 N30 S10 | (R) (S) (R) | — | *aac(3)-IIa* | *aadA16*, *aph(3″)-Ib*, *aph(6)-Id*, *aac(6′)-Ib-cr* |
| Beta-lactam | AMP10 XNL30 | (R) (R) | *bla*<sub>SHV-106/28</sub> | *bla*<sub>TEM-1β</sub> *bla*<sub>SCO-1</sub>, *bla*<sub>CTX-M-15</sub> | *bla*<sub>TEM-1β</sub> |
| Folate pathway antagonist | SXT25 | (R) | — | — | *drfA27*, *sul1*, *sul2* |
| Fluoroquinolone | CIP5 | (R) | *oqxA*, *oqxB* | — | — |
| Macrolide | not tested | | — | — | — |
| Phenicol | C30 | (R) | — | — | *catA2* |
| Rifamycin | not tested | | — | — | *aar3* |
| Tetracycline | D30 T30 | (R) (R) | — | — | *tet(D)* |
| Quaternary ammonium compounds | not tested | | — | — | *qacEDelta* |

\* Antibiotics Disc diffusion panel: G10, gentamycin; N30, neomycin; S10, streptomycin; AMP10, ampicillin; XNL30, ceftiofur; SXT25, sulfamethoxazole-trimethoprim; CIP5, ciprofloxacin; C30, chloramphenicol; D30, doxycycline; T30, tetracycline; —, No predicted resistance gene detected. \*\* ARGs predicted by ResFinder and CARD (perfect and strict).

### 3.3. Correspondence between Antibiotic Resistance Phenotypes and Predicted ARGs

The multidrug resistance profiles of both bacterial isolates were characterized by the Kirby-Bauer disc diffusion assay, which we use routinely to screen for resistance to six classes of antibiotics: (1) aminoglycosides (gentamycin, neomycin, streptomycin), (2) beta-lactams (ampicillin, ceftiofur), (3) tetracyclines (doxycycline, tetracycline), (4) phenicol (chloramphenicol), (5) folate pathway antagonists (sulfamethoxazole-trimethoprim), and (6) quinolone (ciprofloxacin) [13]. *E. coli* HUM3355 and *K. pneumoniae* HUM7199 were selected for this study because they were isolated from human fecal samples and found to be resistant to 10 out of 11 antibiotics tested (Tables 1 and 2). *E. coli* HUM3355 was sensitive only to ciprofloxacin whereas *K. pneumoniae* HUM7199 was only sensitive to neomycin.

Both genomes contained sufficient antibiotic resistance genes to account for each resistance phenotype (Tables 1 and 2). The large number of predicted resistance genes in *E. coli* HUM3355 were distributed across the chromosome and plasmids. In fact, the chromosome encoded genes predicted to confer resistance to all six classes of antimicrobial classes tested. Plasmid pHUM3355_A encoded 21 predicted resistance genes, also covering all classes of antibiotics tested. Fluoroquinolones present an interesting case for efforts to predict phenotypes from gene annotations: achieving clinical resistance to fluoroquinolones requires the additive activities of multiple resistance genes and usually requires a chromosomal resistance allele [40]. Thus, the four genes in *E. coli* HUM3355 that can contribute to fluoroquinolone resistance can be interpreted to potentiate resistance but likely only reduce sensitivity to ciprofloxacin (Table 1). *K. pneumoniae* HUM7199 was found to encode far fewer resistance genes on its chromosome. In this strain, the smaller plasmid pHUM7199_B encoded most resistance genes.

### 3.4. Plasmid Architecture

Three plasmids pHUM3355_A (168,684 bp), pHUM3355_B (155,161 bp), and pHUM3355_C (96,053 bp) from *E. coli* HUM3355 and two plasmids pHUM7199_A (231,078 bp) and pHUM7199_B (94,541 bp) from *K. pneumoniae* HUM7199 were detected. Each of the five plasmids identified in this study were circularized in the assembly workflow, consistent with the dominance of circular plasmids in Enterobacteriaceae. Annotated features are listed in Table S1. Each plasmid belongs to a different incompatibility group and each is a mosaic of gene islands (Figure 3). Four plasmids, pHUM3355_A (*IncA/C2*), pHUM3355_B (*IncFII/IncFIB/IncQ1*), pHUM3355_C (*IncI1α*), and pHUM7199_A (*IncFIB(K)/IncFII(K)*) are predicted to be conjugative since they carry complete conjugative transfer modules (relaxase, Type 4 Coupling Protein, and Type 4 Secretion System-Mating Pair Formation) in the following MOB relaxase family categories: pHUM3355_A (MOBH), pHUM3355_B (MOBF), pHUM3355_C (MOBP1), and pHUM7199_A (MOBF). Only pHUM7199_B (*IncR, IncFIA(HI1)*) is classified as non-conjugative as it carries neither conjugative nor mobilization genes.

The COPLA universal plasmid classification scheme classified the plasmids as follows. pHUM3355_A (PTU-C, host range V), pHUM3355_B (PTU-FE, host range III), pHUM3355_C (PTU-I1, host range III), pHUM7199_A (PTU-FK, host range III), and pHUM7199_B (PTU-R, host range I). Host range V suggests that pHUM3355_A has a very broad host range and can transfer between bacterial Classes. pHUM3355_B, pHUM3355_C, and pHUM7199_A host range III suggests an ability to transfer between bacterial Families. A host range prediction of I suggests pHUM7199_B may be restricted to transfer within the species *K. pneumoniae*.

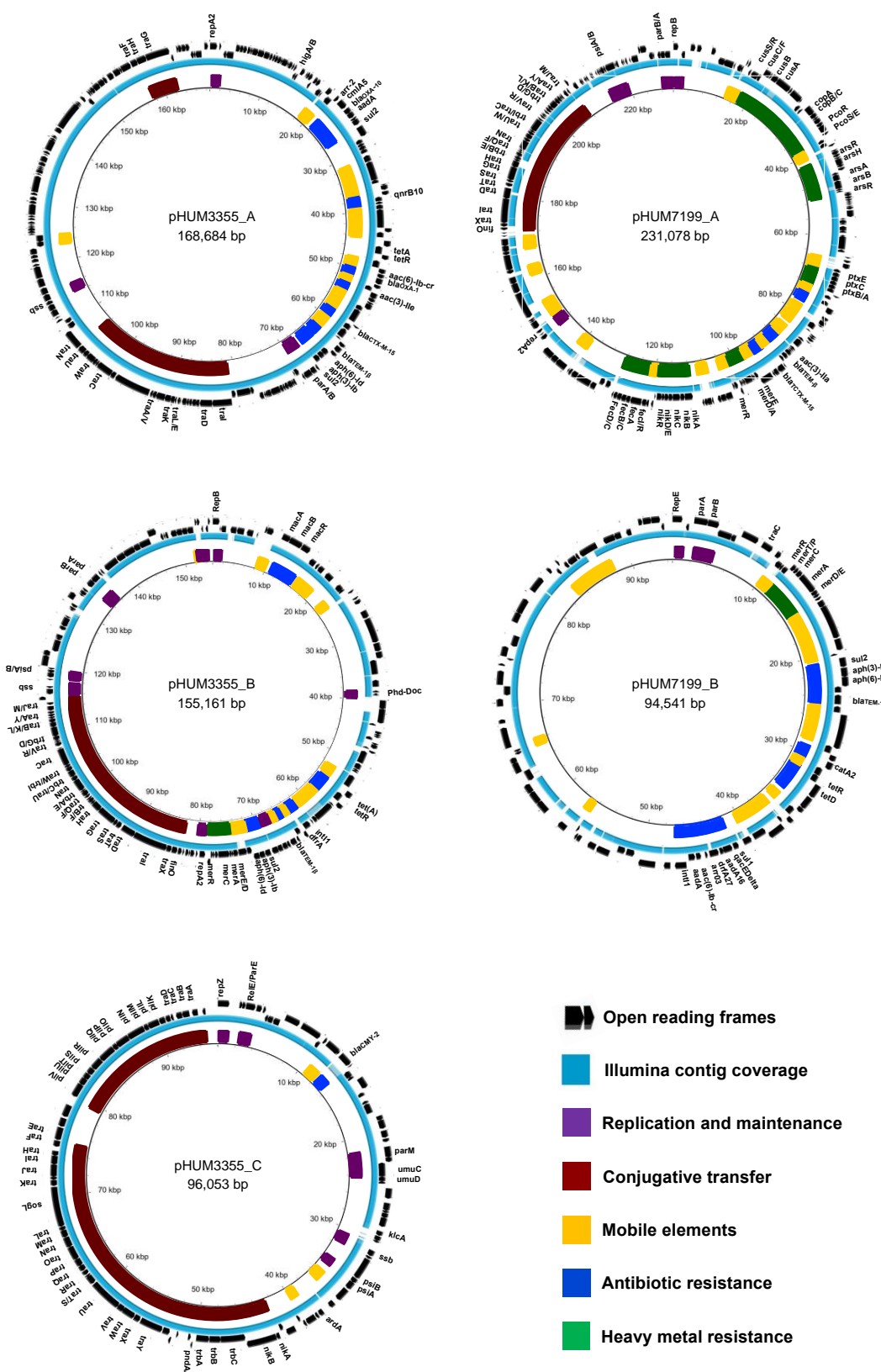

**Figure 3.** Gene maps and annotations for the five plasmids. The annotated functions of genes are indicated with coloured blocks (inner ring), and the regions covered by contigs generated by de novo assembly of Illumina sequencing reads are shown in light blue (middle ring). The outer ring shows predicted open reading frames.

All five plasmids carry maintenance and stability genes associated with either partitioning, addiction, anti-restriction, or SOS inhibition mechanisms (Table S1). Three plasmids carry operons encoding genes associated with heavy metals: mercury (*merE/D/S/C/R*), copper (*cusS/R/C/F/B/A, copA/B/C, PcoR/S/E*), arsenic (*arsR/H*), nickel (*nik*A/B/C/D/E/R) and iron (*fec*I/R/A/B/C/D) (Figure 3).

Comparing the five plasmids to genome sequences available at NCBI GenBank revealed that each plasmid is a newly described genetic element. A large 47,304 bp island in pHUM3355_A carries seventeen resistance genes, including four aminoglycoside (*aac(6)-Ib-cr, aadA, aph(3)-Ib, aph(6)-Ic*), four beta-lactam (*bla*$_{TEM-1\beta}$, *bla*$_{CTX-M-15}$, *bla*$_{OXA-1}$, *bla*$_{OXA-10}$), sulfonamide (*sul2*) phenicol (*cmlA*), quinolone (*qnrB*), rifamycin (*arr-2*), tetracycline (*tetA/tetR*) and a small multi-drug efflux pump gene (*qacEDelta*) conferring resistance to quaternary ammonium compounds. This ARG island is a module unique to pHUM3355_A, though some of the island's gene content overlaps with pHUM3355_B (Table 1).

pHUM3355_B aligned to a nearly identical plasmid sequence pCFS3313-1 (155,171 bp; accession CP026940), which was isolated from *E. coli* from a cow in Ireland. pHUM3355_B contains two clusters of mobile elements and resistance genes. A 15,151 bp region contains a macrolide resistance operon (*macA/B/R*) surrounded by transposable and insertion sequence elements. A second ARG module of 26,232 bp consists of nine ARGs, including tetracycline efflux (*tetA*), trimethoprim (*drfA*), beta-lactam (*bla*$_{TEM-1\beta}$), sulfonamide, two aminoglycoside and mercury resistance operon (*merE/D/A/C/P/T/R*), and a class 1 integron (*intI1*) element. This region also shows high conservation to plasmid sequences CP071134.1, CP026940.2, MH195200.1 and CP018994.1.

pHUM3355_C aligned to plasmid pRHB17-C09_3 (96,066 bp; accession CP057698) from *E. coli* from a pig fecal sample in the United Kingdom and plasmid p12-4374_96 (96,042 bp; accession CP012929) from *Salmonella* Heidelberg from human stool isolated in Canada (Figure 4A). Comparison of pHUM3355_C to other plasmids revealed that the β-lactamase gene *bla*$_{CMY-2}$ is located within a 3649 bp accessory region.

*K. pneumoniae* pHUM7199_A contains one cluster of mobile elements and heavy metal resistance that is highly conserved across the top 10 BLAST hits in GenBank for this plasmid (Figure 4B). This region is composed of mobile elements with genes associated with metals copper and arsenic (*cusS/R/C/F/B/A, copB/C/D/R/S/E, arsR/H/D/A/B*). The second resistance gene cluster is an accessory region (63,308 bp) that is not conserved in the top 10 most closely related plasmid sequences. This region comprises several genes conferring resistance to beta-lactams (*bla*$_{SCO-1}$, *bla*$_{TEM-1\beta}$, *bla*$_{CTX-M-15}$) and genes associated with heavy metals mercury, nickel and iron.

pHUM7199_B has a large 36,410 multi-resistance region that is shared with three other plasmids, all isolated from *K. pneumoniae*: CP009116 from human urine in the USA, and two larger unnamed plasmids CP063009 from human tissue in Russia and CP063012 from human blood in Russia. This region includes the class-1 integron-integrase and multiple resistance gene clusters; [*intI1-acc(6)-Ib-cr-aar-3-drfA27-aadA16-qacEdelta-sul1*], [*tetD-tetR-catA2*], [*blaTEM-1β-aph(6)-Id-aph(3)-Ib-sul2*] and [*merE/D/A/C/P/T/R*] (Figure 4B).

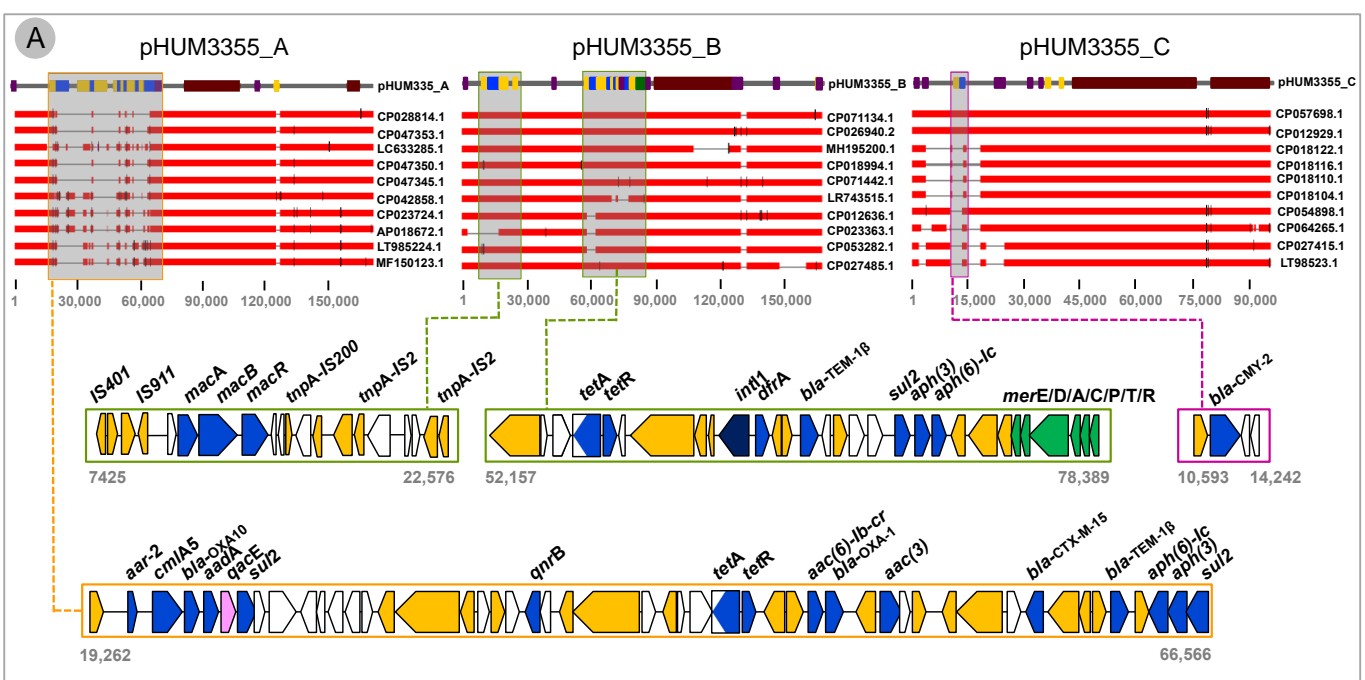

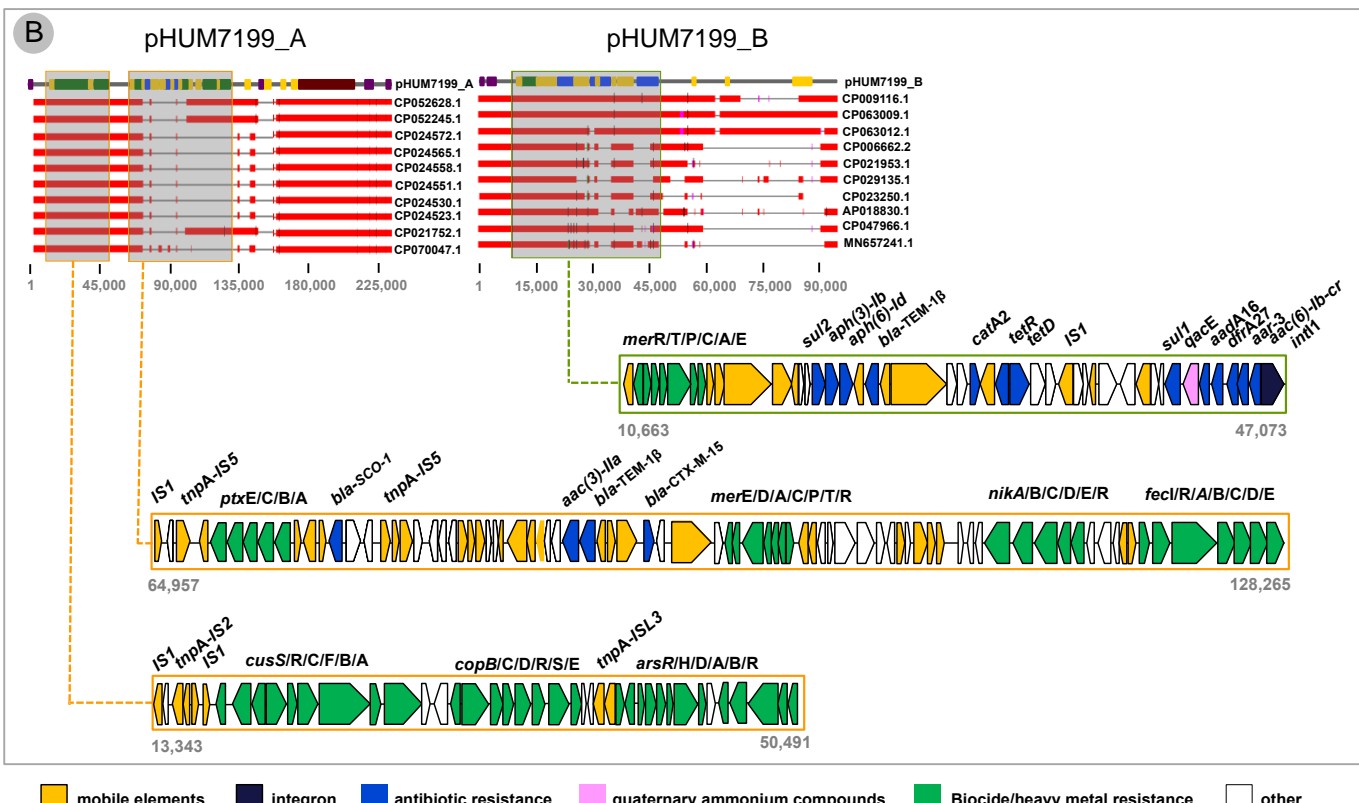

**Figure 4.** Comparative BLAST analysis and genetic context of antimicrobial resistance genes. (**A**) HUM3355 plasmids, and (**B**) HUM7199 plasmids.

## 4. Discussion

Conjugative plasmids carrying clinically relevant ARGs are central drivers of the horizontal transfer of AMR between bacteria, both in nature and in clinical environments [3,41]. Identification of acquired ARGs and associated mobile genetic elements is an important

step in understanding the mechanisms and epidemiology of antimicrobial resistance [36]. Resistance phenotypes alone are insufficient for pathogen identification and source tracking during an outbreak because of extensive overlap in the resistance phenotypes between unrelated bacteria. Equally complicating is that the genetic determinants of resistance can transfer horizontally from reservoirs in non-pathogenic bacteria to pathogens as well as transfer between pathogens.

The 'high accuracy calling' model for Oxford Nanopore Technologies MinION sequencing exceeded our expectations and past experience by generating closed, high quality genome assemblies. Previously we have relied on the high accuracy of Illumina sequencing to correct the relatively low accuracy of Nanopore sequencing; however, with the high accuracy Nanopore assembly, polishing provided only minor refinements. Both genomes were reduced in size by polishing, but gene annotation improved as observed with an increase in perfect matches to ARGs in the CARD database (Figures 1 and 2). Despite the relatively low genome coverage in the Illumina datasets, CARD identified many of the same genes detected in the polished genomes, but some of these were only strict, not perfect, matches to the CARD reference genes (Figure 2). Nanopore sequencing demonstrated the additional capability of resolving multi-copy TEM-1 in both HUM3355 and HUM7199, whereas Illumina only assemblies cannot distinguish multiple identical copies (Figure 2).

The 'One Health' paradigm recognizes the mobility of ARGs through interconnected human, animal and environmental systems [42–46]. Our study emphasizes the importance of One Health considerations for combating AMR. In two extensively drug resistant bacteria isolated from human diarrhoea in a small town in northern Botswana, we sequenced and assembled five antimicrobial resistance plasmids that contain gene islands previously identified in human and animal systems on other continents. Two plasmids, pHUM3355_B and pHUM3355_C are almost identical to plasmids isolated independently from humans and pigs in North America and Europe. With few comparator sequences available it is impossible to infer where and when the common ancestors of these plasmids originated or how they became co-resident in *E. coli* in northern Botswana. The Chobe district is a low traffic but highly prized tourist destination for Europeans and Americans, prompting us to speculate that tourism has transported these resistance plasmids into and/or out of the Chobe district to other corners of the planet.

LMICs will continue to face the greatest burdens of infectious diseases and AMR. The democratization of DNA sequencing through rapid technological advances attracts much attention. Yet, DNA sequencing is only one component of genomic-informed infectious disease epidemiology. A need for computer language and coding abilities to execute the more popular DNA assembly packages makes bioinformatics a hurdle in public health and research labs around the world. Free-to-use web-based platforms are effectively reducing barriers by providing popular workflows in graphical user interfaces. We used the Centre for Genomic Epidemiology (https://www.genomicepidemiology.org, accessed on 1 July 2019) extensively for its multiple predictive tools. The Bacterial and Viral Bioinformatics Resource Centre (www.BV-BRC.org) provides high quality and easy to use workflows for genome assembly and annotation using Illumina or Oxford Nanopore Technologies DNA sequence data. Additionally, restrictive in LMICs is the need for sufficient computer infrastructure; fortunately, assembly tools can be launched on most desktop computers and platforms like BV-BRC provide the necessary computational capacity.

In this study we detected plasmid-borne AMR genes in both *E coli* HUM3355 and *K. pneumoniae* HUM7199 associated with resistance to seven clinically relevant antimicrobial classes: aminoglycoside, beta-lactam, chloramphenicol, rifamycin, sulfonamide, trimethoprim, tetracycline, and quaternary ammonium compounds. Additionally, plasmids pHUM3355_B and pHUM7199_B were found to carry class 1 integron elements, which are gene capture elements associated with global dissemination of ARGs [47]. All three plasmids identified in *E. coli* were predicted to be self-transmissible and belong to different incompatibility groups. pHUM3355_A belongs to the IncC incompatibility group, a large group of broad-host range conjugative plasmids that are the main drivers of global

dissemination antibiotic resistance among several pathogenic species of *Gammaproteobacteria* associated with food products, food-producing animals, and humans [48]. The top 100 blast hits (>60% Query Coverage, >99% Identity) from query sequence of pHUM3355_A correspond to resistance plasmids isolated from a wide range of genera within phylum Proteobacteria, including *Edwardsiella, Proteus, Klebsiella, Escherichia, Providencia, Enterobacter, Vibrio, Yersinia, Salmonella, Aeromonas,* and *Photobacterium*. Plasmid pHUM3355_B corresponds to plasmids in *Escherichia* and *Salmonella*, with closely related plasmids identified in pathogenic strains of *E. coli* in the UK [49] and *S. enterica* serovar Typhimurium in Ghana [50]. PlasmidFinder identified a truncated IncQ replicon region in pHUM3355_B; the 529 bp match to the full length (796 bp) replicon is annotated as non-functional according to PlasmidFinder. Plasmid pHUM3355_C is a typical IncI1α plasmid, containing highly conserved *tra* and *pil* conjugative transfer regions. Similar plasmids carrying the signature beta-lactamase CMY-type gene have been described globally [51]. The top blast hits from query pHUM3355_C correspond to plasmids circulating in *Escherichia, Shigella,* and *Salmonella*.

The multi drug resistant *K. pneumoniae* HUM7199 carried one predicted conjugative plasmid (pHUM7199_A, IncFIB(K)/IncFII(K)) and one non-mobile plasmid (pHUM7199_B, IncR). The IncFIB (also denoted IncFIB$_K$) is known to be widespread among *Klebsiella* strains playing a role in the dissemination of antimicrobial resistance and virulence genes [52]. Top BLAST hits of the query pHUM7199_A are exclusively found in *K. pneumoniae*, potentially reflecting the high number of plasmids that have been sequenced in *Klebsiella*. The IncR plasmid replicon was first discovered in 2006 in multi-drug resistant *K. pneumoniae* plasmid pK245 [53], and this plasmid type is observed to allow co-selection of both virulence and resistance functions [54]. pHUM7199_B shares a genetic backbone composition related to other previously described IncR plasmids, which also appear to be non-conjugative [53–55].

## 5. Conclusions

Plasmids are key epidemiological markers for tracking the emergence and dissemination of clinically relevant ARGs in human, animal, and ecosystem, but they are largely overlooked because most projects use Illumina's short-read DNA sequencing, which is insufficient for assembling most plasmids. Combining Illumina and Nanopore sequencing approaches proved superior for accurate resolution of whole genome architecture and improved ARG calling. As Botswana and other African nations begin to apply whole genome sequencing of pathogens for public health, this study validates the benefits of combined short- and long-read DNA sequencing data, but also demonstrates that the advances in Oxford Nanopore Technologies sequencing can provide robust assemblies and high-confidence detection of ARGs even in the absence of supporting Illumina data.

**Supplementary Materials:** The following supporting information can be downloaded at: https://www.mdpi.com/article/10.3390/applmicrobiol2040071/s1, Figure S1: Short-read and long-read assembly workflows and Table S1: Comparative genomics of pHUM3355 and pHUM7199 plasmids incompatibility (*Inc*), genes associated with replication, conjugative transfer and maintenance/stability.

**Author Contributions:** All authors contributed to the research project. A.D.S.C. and K.A.A. designed the research and acquired funding. T.O.R., A.D.S.C., N.A.L. and S.V.B. conducted the research. T.O.R. and A.D.S.C. wrote the manuscript and all authors contributed to manuscript review and editing. Conceptualization, A.D.S.C. and K.A.A.; formal analysis, T.O.R., A.D.S.C., N.A.L. and S.V.B.; investigation, T.O.R., A.D.S.C., N.A.L. and S.V.B.; writing—original draft preparation, T.O.R. and A.D.S.C.; writing—review and editing, T.O.R., A.D.S.C., N.A.L., S.V.B. and K.A.A.; funding acquisition, A.D.S.C. and K.A.A. All authors have read and agreed to the published version of the manuscript.

**Funding:** This work is based on the research supported by a QEII Research Scholar award from Universities Canada and the University of Regina to A.D.S.C., a Discovery grant RGPIN-2019-07135 from the Natural Sciences and Engineering Research Council of Canada to A.D.S.C., a Postdoctoral Research Fellowship from the Saskatchewan Health Research Foundation to S.B., and the National Science Foundation Dynamics of Coupled Natural and Human Systems Award 1518486 to K.A.A. The funders had no role in study design, data collection and analysis, decision to publish, or preparation of the manuscript.

**Data Availability Statement:** The sequence data from this study is available at the National Center for Biotechnology Information (NCBI) Sequence Read Archive (SRA) as BioProject PRJNA810075. The assembled genetic elements are available at GenBank with the following accession numbers: *Klebsiella pneumoniae* HUM7199 (CP093314, CP093315, CP093316) and *Escherichia coli* HUM3355 (CP093317, CP093318, CP093319, CP093320).

**Acknowledgments:** We thank the Center for African Resources: Animals, Communities and Land Use (CARACAL), Botswana for providing the bacterial isolates and phenotypic typing. This research was enabled in part by computing and software support provided by Simon Fraser University's Cedar computing system and the Digital Research Alliance of Canada (alliancecan.ca).

**Conflicts of Interest:** The authors declare that there is no conflict of interest regarding this manuscript.

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
