# Peer review of "Globally Disseminated Multidrug Resistance Plasmids Revealed by Complete Assembly of Multidrug Resistant Escherichia coli and Klebsiella pneumoniae Genomes from Diarrheal Disease in Botswana"

_2673-8007, doi:10.3390/applmicrobiol2040071_

Round 1

Reviewer 1 Report (New Reviewer)

Rahube et al assessed long-read Nanopore and short-read Illumina platforms alone or in combination for laboratory diagnostics in Botswana by annotating plasmids and extensively drug resistant E. coli and Klebsiella. Although, the authors performed a rigorous analysis of the sequence; the results and discussion sections were lacking the comparisons of Illumina and Nanopore platforms. Therefore, it would be better to expand those sections further to discuss the comparisons more. 

In addition, following questions may be addressed:

Is one platform better than other?

Would using just one platform enough for the routine analysis? If so, which one?

What were the pros/cons of each platform in your analysis?

Author Response

Reviewer 2 Report (New Reviewer)

Rahube et al. isolated two multidrug resistant bacterial strains from human infections in the Chobe district in Northern Botswana. They sequenced these two strains using both MiSeq and Nanopore. They further assembled the reads into chromosomes and plasmids. By searching against CARD, they identified ARGs in the two genomes. My major criticism is that the manuscript is sloppy in writing and data analysis. 

Works in a few groups—Alvaro San Millan, Anton Nekrutenko, and Tal Dagan—are quite important in plasmid evolution. However, they were not discussed in the introduction. 

Major comments: 

1. This is essentially a bacterial genome sequencing and assembly manuscript. So make the data analysis process accessible and reproducible. The authors also mentioned "lack of bioinformatics capacity" in line 76. So use Galaxy (https://usegalaxy.org/) to analyze the data and publish the data analysis histories. The Galaxy community even pushed a paper out assembling bacterial genomes using MiSeq and Nanopore reads (https://doi.org/10.1101/347625). 

2. Following on 1, some of the plasmids are really large. I need to see the data analysis workflow and run it myself to make sure these are extrachromosomal plasmids indeed.  

3. Following on 1 and 2, the authors claimed four plasmids as conjugative and suggested that conjugation should be responsible for the dissemination of the AB-resistant genes. I can see the tra genes, but do these four plasmids carry oriT? 

4. In line 56, the authors claimed that "Plasmids evolve and change much faster than their hosts". Well, this is not necessarily true. With increasing selective pressure, plasmid variant(s) can quickly sweep a population (https://doi.org/10.1038/s41559-016-0010). However, under constant selective pressure, plasmids can first acquire variants at very low frequencies as a response to the pressure, and quickly revert to wild-type because of interference events at the chromosomal level (https://doi.org/10.1093/gbe/evz197). Actually, the per plasmid mutation rate is lower compared to the per chromosome mutation rate (https://doi.org/10.1093/gbe/evz197). 

5. Another reason that could lead to slower plasmid evolution is segregational drift. As suggested in (https://doi.org/10.1093/molbev/msy225), "mutations that emerge on multicopy plasmid genomes are repeatedly “diluted” in the population due to segregational drift." The Dagan group has another paper (https://doi.org/10.1093/molbev/msab283), which needs to be discussed as well. 

6. Read carefully the recent papers from these three groups. Be sure to discuss relevant papers properly. 

Other major comments: 

The "2. Materials and Methods" section is sloppy in describing the methods. Try your best to make sure the readers can re-do this work when they have the same strains. So, reproducibility please.  

Minor comments: 

1. In "2.1 Bacterial isolation and antibiotic susceptibility testing", the measure of AB is not clear. Use working concentrations (i.e., µg/mL). Also, specify the supplier and catalog number for each AB. 

2. Line 118, "miniprep kit", add catalog number. 

3. Line 120, "300 bp paired-end reads" → "300 x 300-nt paired-end reads". 

4. Line 126, "basecalled and demultiplexed" → "demultiplexed and basecalled". 

5. I am not sure I followed very well about lines 197–199, "For the HUM7199 genome, polishing by Pilon reduced the overall predicted genome size. The HUM7199 chromosome acquired 1,563 bp whereas the second largest element shrunk after polishing the Nanopore assembly with Illumina."

Round 2

Reviewer 2 Report (New Reviewer)

Thank you for making the revisions. My comments have been addressed. I have no further questions.

This manuscript is a resubmission of an earlier submission. The following is a list of the peer review reports and author responses from that submission.

Round 1

Reviewer 1 Report

I confirm that I have reviewed a manuscript titled "Globally disseminated multidrug resistance plasmids in E. coli and Klebsiella from diarrheal disease in Botswana. The paper is well written but I have detected some flaw which must be addressed.

Comments

The title must be revised and reads as "Globally disseminated multidrug resistance plasmids in Escherichia coli and Klebsiella pneumoniae species from diarrheal disease in Botswana."

Table 1 and 2: How did the authors determine the resistance profiles of your isolates? Which CLSI version did the authors use?

Line 14: insert "be" between " the words  "can and transfer."

Line 32: Delete "Botswana" from the keywods .

Line 77: write labs in full "laboratories".

Lines 84-93: The lines can be deleted.

Line 97: Start the sentence with " The" instead of "These"

Line 104: Replace a word "presenting" with "admitted"

Line 105: Put the reference.

Line 107: fecal.

Line 108: How did you know that those antibiotics are commonly used in Botswana? Did you use questionnaire?.

Lines 117-118: How many samples (isolates) did you use for DNA extraction? The authors must specify.

Line 142: This not the correct figure referenced.

Lines 191-199: This part can be moved to the discussion section.

Lines 102-104: Delete these lines.

Lines 201-229: The authors must include the results for "Whole genome comparative analysis" as stated under subheading 3.2.

Line 244: How did the authors  use Kirby-Bauer disc diffusion to identify the isolates?

Lines 338-340: These lines must be moved to the discussion.

The conclusion must be revised to support the results.

Reviewer 2 Report

Rahube et al. describe the investigation using whole genome sequencing of plasmids carrying antimicrobial resistance genes from E. coli and Klebsiella species obtained in Botswana. The topic of the manuscript is interesting, but chaotic organization and lack of essential data make it very hard to percept. I have the following comments.

 Major comments:

1. The main goal of the article was set as “to compare assembly and annotation work-82 flows using Illumina and Oxford Nanopore sequence data”. It would indeed be very interesting to compare the plasmid structure predictions using short and long-read sequencing. However, all comparisons in the manuscript are given between short-read assembly and hybrid short- and long-read assembly. It is evident that combination of data provides more information, and hybrid assemblies are much more accurate – but it is not scientifically sound to compare such assemblies since both of them include the short read data. I strongly suggest making additional comparison between short-read only and long-read only assemblies.

2. Only two plasmids are chosen – such a comparison cannot show real differences in two sequencing approaches. The discussion is focused on the plasmids themselves and does not provide any description of sequencing approach differences. It is not clear what was the main goal of the article – either description of two plasmids or making comparison between their assemblies. If the plasmids are in fact very similar to the ones from a database, then they could be assembled using short reads only by using reference-aware assembly – it is not clear what added value was obtained by long-read sequencing in this case.

3. Plasmids cannot be assigned to more than one incompatibility classes by definition. The data obtained using PlasmidFinder (were there 100% identity for each replicon signature? I doubt this) should be always additionally verified using, e.g., comparison with reference plasmid sequences from Genbank. For example, you predicted that pHUM3355_B could have IncFII/IncFIB/IncQ1 type – but IncQ1 is known to usually have MOBQ relaxase family and is non-conjugative, while your prediction was MOBF and conjugation ability. You should verify your data to set only one incompatibility type for each plasmid, otherwise your data will be senseless. This is a very serious concern since the presence of different replicon signatures in an assembled plasmid may indicate the errors in its assembly, e.g., mixture of several plasmids into one.

Minor comments:

Please provide a full name for E. coli in the title. I also suggest adding the application of WGS and/or workflow comparison to the title since they are key points of the manuscript and their absence in a title could be misleading for potential readers. In addition, I suggest adding species name “Klebsiella pneumoniae” since only this species was studied, not the whole genera.

Line 96 – the term “extensively multi-drug resistant” is misleading. The bacteria are usually classified as multidrug-resistant (to 3 or more classes of antimicrobials, MDR) and extensively drug resistant (to all but one or two classes, XDR). Later you describe the isolates as multidrug-resistant (line 164). Please choose the correct option.

Lines 99-102 – there is a special place for ethical committee approval at the end of the manuscript (please refer to author instructions on journal website)

Line 113 – why the antibiotic testing lacks most cephalosporins and carbapenems? Are they not used in Botswana?   

Line 127 – the depth of 10x is usually considered insufficient for precise genome sequencing (30-40 x being usually used). Why the coverage was so low?

Line 142 – figure 1 does not show anything regarding genome annotation. Please fix or remove this statement.  

Line 177 – you first claim that the assemblies were obtained using “Nanopore data alone” referring to figure 1b and then add “Nanopore Assembly polished with Illumina” in the Figure itself. These claims contradict each other. What is correct?

Please add species names, not only strain codes, to the headers of tables 1 and 2 – this will greatly increase readability

Line 293 – ‘Classes’ does not need to be capitalized

Line 303 – “each plasmid is a newly described genetic element” – what do you mean by this? Later you stated that the plasmid “aligned to a nearly identical plasmid sequence” in Genbank.